# A Novel Ageladine A Derivative Acts as a STAT3 Inhibitor and Exhibits Potential Antitumor Effects

**DOI:** 10.3390/ijms24108859

**Published:** 2023-05-16

**Authors:** Na He, Li Li, Rui Li, Si-Qi Zhang, Li-Hong Wu, Xian Guan, Qian-Yue Zhang, Tao Jiang, Jin-Bo Yang

**Affiliations:** 1Key Laboratory of Marine Drugs, Ministry of Education, School of Medicine and Pharmacy, Ocean University of China, Qingdao 266003, China; hena@stu.ouc.edu.cn (N.H.); lili3950@stu.ouc.edu.cn (L.L.); lirui02@luye.com (R.L.); zsq@ouc.edu.cn (S.-Q.Z.); wulihong@stu.ouc.edu.cn (L.-H.W.); 12218126@zju.edu.cn (X.G.); zqy501@stu.ouc.edu.cn (Q.-Y.Z.); 2Innovation Platform of Marine Drug Screening & Evaluation, Qingdao National Laboratory for Marine Science and Technology, Qingdao 266100, China; 3Laboratory for Marine Drugs and Bioproducts, Qingdao 266237, China

**Keywords:** JAK/STAT3 signaling pathway, STAT3 inhibitors, ageladine A derivatives, dibromo pyrrole-imidazole, antitumor mechanisms, antitumor activities

## Abstract

The Janus kinase/signal transducer and activator of the transcription 3 (JAK/STAT3) signaling pathway controls multiple biological processes, including cell survival, proliferation, and differentiation. Abnormally activated STAT3 signaling promotes tumor cell growth, proliferation, and survival, as well as tumor invasion, angiogenesis, and immunosuppression. Hence, JAK/STAT3 signaling has been considered a promising target for antitumor therapy. In this study, a number of ageladine A derivative compounds were synthesized. The most effective of these was found to be compound **25**. Our results indicated that compound **25** had the greatest inhibitory effect on the STAT3 luciferase gene reporter. Molecular docking results showed that compound **25** could dock into the STAT3 SH2 structural domain. Western blot assays demonstrated that compound **25** selectively inhibited the phosphorylation of STAT3 on the Tyr705 residue, thereby reducing STAT3 downstream gene expression without affecting the expression of the upstream proteins, p-STAT1 and p-STAT5. Compound **25** also suppressed the proliferation and migration of A549 and DU145 cells. Finally, in vivo research revealed that 10 mg/kg of compound **25** effectively inhibited the growth of A549 xenograft tumors with persistent STAT3 activation without causing significant weight loss. These results clearly indicate that compound **25** could be a potential antitumor agent by inhibiting STAT3 activation.

## 1. Introduction

The JAK/STAT3 signaling pathway plays crucial roles in many physiological processes, including modulating the proliferation, differentiation, and apoptosis of cells [1]. In healthy cells, STAT3 is typically located in the cytoplasm as an inactive dimer and is strictly mediated [2]. However, sustained activation of STAT3 causes various diseases, such as rheumatoid arthritis, atherosclerosis, stroke, myocardial ischemic injury, and cancer [3]. Overactivated STAT3 signaling contributes to malignant progression and poor prognosis by promoting the proliferation, survival, metastasis, and invasion of cancer cell, as well as angiogenesis and immune evasion [4]. It has been reported that aberrantly activated STAT3 is present in nearly 70% of human cancer types, including colorectal, lung, breast, prostate, liver, pancreatic, multiple myeloma, and leukemia [5,6]. Thus, STAT3 signaling has been recognized as a novel anticancer target [7], motivating researchers to identify and develop effective STAT3 inhibitors.

Recently, scientists have discovered a number of indirect and direct STAT3 inhibitors [8,9]. Indirect inhibitors target upstream molecules in the STAT3 signaling pathway, such as JAK and Src, which diminish STAT3 phosphorylation and impact the expression of downstream proteins [10]. Several JAK inhibitors are currently in clinical trials or have already been approved for clinical treatment, such as sorafenib and ruxolitinib [11]. However, the lack of specificity in indirect STAT3 inhibitors may result in some negative effects due to the crucial roles played by upstream molecules in normal physiological processes [12]. In contrast, direct STAT3 inhibitors interact with STAT3, potentially alleviating the side effects associated with indirect inhibitors.

There are three main categories of STAT3 direct inhibitors: molecular probes, DNA-binding domain (DBD) molecule inhibitors, and small-molecule SH2 domain inhibitors [13]. Several compounds have been identified as STAT3 direct inhibitors; however, only a few of these have entered preclinical trials, such as pyrimethamine [14], STA-21 [15], HJC0416 [16], TTI-101 [17], WP1066 [18], and BBI608 [19]. However, none of the STAT3 inhibitors have yet been approved for cancer therapy. Therefore, it is imperative to develop STAT3 inhibitors with enhanced pharmacological properties and potency.

Natural marine compounds with distinctive structures and extensive halogen modifications are crucial sources of drugs, many of which exhibit antitumor activity [20]. Ageladine A is a hydrophilic dibromopyrrole-imidazole alkaloid first isolated by Japanese scientists from ageladine sponges in 2003 [21]. Ageladine A exhibits flavin-like fluorescence and membrane permeability [22]. In biofunctional studies, ageladine A has been found to inhibit matrix metalloproteinases (MMPs) at the micromolar level, particularly MMP2 [23]. To date, most studies of ageladine A and its derivatives have focused on their synthesis; only a few have considered their pharmacological aspects, and further exploration of their pharmacological mechanisms and action targets is now required [24].

For this reason, we carried out a high-throughput screening of certain natural marine compounds and their derivatives from our in-house library using a STAT3-dependent reporter. We found that the ageladine A derivative compound **25** strongly inhibited luciferase activity (Appendix A). In brief, in this study, we preliminarily evaluated the antitumor activity of ageladine A and its derivatives, explored the pharmacological mechanisms of action with compound **25**, and performed in vitro and in vivo antitumor experiments.

## 2. Results

All compounds were synthesized and characterized as described in the Appendix A, which shown in Appendix A. The liquid purity analysis of compound **25** is shown in Appendix A.

### 2.1. Compound ***25*** Inhibited STAT3-Based Luciferase Activity and Tumor Cell Growth

To evaluate the inhibitory activity of ageladine A and its derivatives on the STAT3 signaling pathway, the luciferase-expressing cell line SKA based on constitutive STAT3 activation was selected for screening [25]. Initially, ageladine A and its derivatives **11**–**28** were prepared, and their effects on luciferase activity were measured. Those derivatives that inhibited luciferase activity are shown in Table 1. It can be seen that the ageladine A derivatives **14**, **15**, **25**, and **28** exhibited significant luciferase-inhibitory activity. Subsequently, four cancer cell lines (A549, DU145, Hela, and MDA-MB-231) were used to investigate the antiproliferative activity of **14**, **15**, **25**, and **28** (Table 2). We found that compound **25** exhibited significant antiproliferative activity against these four cancer cells. In addition, we examined the antiproliferation effects of compound **25** on two normal cells, i.e., HUVECs and BMs (mouse-derived bone marrow cells). The results indicated that compound **25** had low toxicity to normal cells (Table 3). In light of these results, we continued our study using compound **25** only.

### 2.2. Compound ***25*** Bound Directly to STAT3 SH2 Domain

To investigate the binding mode of compound **25** with the target protein STAT3, we employed MOE 2022 (Molecular Operating Environment 2022) docking software to simulate the interactions between compound **25** and STAT3 (PDB: 1BG1). The binding model of compound **25** and the STAT3 protein is shown in Figure 1A. Compound **25** bound to the STAT3 protein and interacted with the amino acid residues Met648 and Arg688 in the STAT3 SH2 structural domain. To confirm the direct interaction between compound **25** and STAT3, surface plasmon resonance (SPR) experiments were carried out. As illustrated in Figure 1B, compound **25** bound to the wild-type STAT3 protein with an equilibrium dissociation constant (K_D_) of 9.01 μM. The STAT3-dependent luciferase reporter assays indicated that compound **25** also inhibited luciferase activity in a dose-dependent manner (Figure 1C). Finally, CETSA experiments were performed to investigate the interaction between compound **25** and the STAT3 protein. As can be seen in Appendix A, STAT3 was more stabilized in **25**-treated cells than in vehicle-treated cells. These results suggested that compound **25** may bind directly to STAT3.

### 2.3. Compound ***25*** Inhibited Constitutive and IL-6-Induced pY705-STAT3

Persistent STAT3 activation results in tumor formation and poor prognosis in case of malignancies [26]. In the current study, we found that compound **25** decreased STAT3 Y705 phosphorylation in a time-dependent manner, but had little effect on total STAT3 (Figure 2A,B). However, compound **25** did inhibit STAT3 Y705 phosphorylation in DU145 and A549 cell lines at a concentration of 5 µM (Figure 2C,D).

Multiple cytokines and growth factors are capable of activating the JAK/STAT3 signaling pathway [27]. In addition, elevated IL-6 levels hyperactivate the JAK/STAT3 signaling pathway, as has been observed in many cases of hematopoietic or solid malignancies [28]. In the current study, to determine whether compound **25** decreases the activation of STAT3 induced by IL-6, we pretreated Hela and MDA-MB-231 cells with different concentrations (0, 2.5, 5, 10, 15 μM) of compound **25** for 2 h; these were then stimulated with 20 ng/mL IL-6 for 20 min. Western blot analysis demonstrated that compound **25** obviously inhibited IL-6-induced STAT3 Y^705^ phosphorylation in Hela and MDA-MB-231 cell at a concentration of 10 µM (Figure 3A,B). In both DU145 and A549 cells, compound **25** consistently downregulated the expression of the downstream proteins c-Myc, Cyclin D1, and Bcl-xL (Figure 3C,D). Taken together, these results indicated that compound **25** inhibited the constitutive and IL-6-induced pY705-STAT3 in cancer cells.

### 2.4. Compound ***25*** Selectively Inhibited STAT3

Specificity is one of the main challenges of STAT3 inhibitors [29]. To verify the selectivity of compound **25**, we investigated its effect on the phosphorylation of JAK. As can be seen in Figure 4A,B, compound **25** had little impact on the phosphorylation of JAK in A549 and DU145 cells. The quantifications of p-JAK/α-tubulin are displayed in Appendix A. Compared with the control group, there were no significant differences in each group. In addition, the expression of p-NF-κB, p-AKT, p-GSK3β, and p-JNK in A549 and DU145 cells did not significantly change at the designated times of compound **25** treatment (Figure 4C,D), indicating that the effects of compound **25** treatment on other signaling pathways were slight over the course of the study period. The Western blot results were then quantified, as shown in Appendix A. As a member of the STAT protein family, STAT3 exhibits definite homology with other STAT proteins. For this reason, we also detected the effect of compound **25** on STAT1 and STAT5 phosphorylation in tumor cells. We found that compound **25** had little effect on the expression of STAT1 and STAT5 phosphorylation in A549 cells (Figure 4E). The Western blot quantification results are shown in Appendix A. These results confirmed that compound **25** could selectively bind to STAT3.

### 2.5. Compound ***25*** Suppressed Proliferation, Survival, and Migration of Cancer Cells

Consistently activated STAT3 enhances tumor growth and metastasis [30]. In the current study, to confirm in vitro antitumor activity, the antiproliferation activity of compound **25** was analyzed. As can be seen in Figure 5A, compound **25** suppressed the proliferation of A549, DU145, Hela, and MDA-MB-231 cells in a concentration-dependent manner, with IC_50_ values of 4.42 ± 0.42 μM, 8.73 ± 1.53 μM, 8.67 ± 0.34 μM, and 5.599 ± 1.36 μM, respectively. Colony survival assays also revealed that compound **25** strongly inhibited colony formation at 5 μM in A549 and DU145 cells (Figure 5B). Furthermore, we also detected an inhibitory effect on tumor cell migration using scratch assays; the obtained data showed that compound **25** obviously repressed the migration of A549 and DU145 cells (Figure 5C,D).

### 2.6. Compound ***25*** Inhibited the Growth of A549 Xenograft Tumors

To further evaluate the antitumor efficacy of compound **25** in vivo, we examined its effect on the A549 xenograft tumor model. As shown in Figure 6A–C, 10 mg/kg compound **25** effectively inhibited the growth of A549 xenograft tumors, compared with the vehicle group, after drug treatment for 14 days. The tumor-inhibitory effect of 10 mg/kg compound **25** was 39.5%. Meanwhile, there were no significant changes in the body weights of mice in each group during treatment (Figure 6D). H&E staining was performed on the livers and kidneys of mice from both groups. The results revealed no significant liver or kidney damages during the treatment period (Figure 6E).

The tumor mitotic index (Ki67) and STAT3 signaling pathway in tumor tissues were further evaluated by immunohistochemistry (IHC). As shown in Figure 7A–C, compound **25** suppressed the expression of Ki67, pY705-STAT3, and c-Myc in A549 xenograft tumors, compared with the vehicle group. These data indicated that compound **25** inhibited A549 xenograft tumor growth and the STAT3 signaling pathway without causing severe toxicity.

## 3. Discussion

STAT3 transmits transcriptional signals to the nucleus through phosphorylation and dimerization, and is tightly regulated in healthy cells [31]. Aberrantly activated STAT3 occurs frequently in multiple cancer types; plays an important role in tumor formation, metastasis, and drug resistance; and is associated with poor clinical prognosis of cancer [32]. In humans, hyperactivated STAT3 affects the development and progression of cancer by promoting tumor invasion, tumor-cell proliferation and survival, angiogenesis, and immunosuppression [33]. Given the prevalence and importance of STAT3 signaling in tumors, targeting the STAT3 signaling pathway has proven to be a promising strategy in the development of antitumor drugs, particularly those that target STAT3 directly and selectively.

In this study, a derivative of the natural marine compound ageladine A, compound **25**, was found to act as a potential STAT3 inhibitor through STAT3-dependent reporters and cell-based screening strategies, which inhibited STAT3 transcriptional activity and cancer cell growth in the micromolar range. Through comparison and analysis, compound **25** was found to be the most effective compound among the investigated derivatives. Compound **25** exhibited stronger inhibitory activity against luciferase-expressing SKA cells with constitutive STAT3 activation, compared with other derivatives. Docking results revealed that compound **25** could interact with amino acid residues in the SH2 domain, such as Met648 and Arg688. We verified the affinity and interaction between compound **25** and the STAT3 protein by SPR and CETSA analysis. Western blot analysis demonstrated that compound **25** inhibited constitutive STAT3 Y705 phosphorylation in A549 and DU145 cell lines, and also inhibited IL-6-induced STAT3 Y705 phosphorylation in Hela and MDA-MB-231 cell lines. Compound **25** had little effect on p-JAK, p-STAT1, p-STAT5, or other signaling pathways, implying that compound **25** selectively reduced STAT3 activation in tumor cells. Furthermore, compound **25** downregulated the expression of the STAT3 downstream proteins c-Myc, Cyclin D1, and Bcl-xL. Compound **25** also suppressed proliferation and migration in both A549 and DU145 cells. In vivo experiments revealed that 10 mg/kg of compound **25** significantly inhibited the growth of A549 xenograft tumors without affecting the body weight of mice.

Taken together, these results suggest that compound **25** acts as a STAT3 inhibitor and exhibits potential antitumor effects. Furthermore, it is critical to investigate the antitumor effects of ageladine A and its derivatives and determine pharmacological mechanisms and action targets, which will be valuable for their future drug development research. In short, more studies are needed to describe the pharmacokinetic characteristics of compound **25** and other derivatives.

## 4. Materials and Methods

### 4.1. Cell Lines and Cell Culture

A549, Hela, MDA-MB-231, BM, and SKA cells (A549 cells transfected with STAT3-driven luciferase reporter gene) were cultivated in Dulbecco’s modified Eagle’s medium (DMEM, Gibco, Grand Island, NY, USA). DU145 cells were cultured in Roswell Park Memorial Institute (RPMI) 1640 medium (Gibco, Grand Island, NY, USA). HUVEC cells were cultured in Ham’s F-12K medium (Gibco, Grand Island, NY, USA). Cell culture media were supplemented with 10% fetal bovine serum (FBS, Gibco, Grand Island, NY, USA), penicillin (100 IU/mL), and streptomycin (100 mg/mL), and all cells were incubated under standard culture conditions at 37 °C in an incubator containing 5% CO_2_. A549, DU145, Hela, MDA-MB-231, and HUVEC cell lines were obtained from the American Type Culture Collection (Manassas, VA, USA).

### 4.2. Antibodies and Reagents

Antibodies against p-Tyr1022/1023-JAK1 (#3331), p-Tyr1007/1008-JAK2 (#3776), p-Tyr980/981-JAK3 (#5031), p-Tyr1054/1055-TYK2 (#9321), p-Tyr701-STAT1 (#9167), p-Tyr705-STAT3 (#9145), p-Tyr694-STAT5 (#9359), JAK1 (#3332S), JAK2 (#3230), JAK3 (#8863), TYK2 (#9312), STAT1 (#14994), STAT3 (#12640), STAT5 (#25656), c-Myc (#13987), cyclin D1 (#AF0931), Bcl-xL (#2764), p-Ser536-NF-κB (#3033), p-Thr308-AKT (#9275), p-Ser9-GSK-3β (#5558), p-Thr183/Tyr185-SAPK/JNK (#4668), NF-κB (#8242), AKT (#4691), GSK-3β (#9832), and SAPK/JNK (#9252) were purchased from Cell Signaling Technology (Danvers, MA, USA). Antibodies against α-tubulin were purchased from Santa Cruz Biotechnology (Santa Cruz, CA, USA). Protease inhibitor (B14001) and phosphatase inhibitor (B15001) were purchased from Bimake (Houston, TX, USA). Recombinant human STAT3 protein (ab268982) was obtained from Abcam (Cambridge, Britain). Recombinant human interleukin-6 (IL-6) protein (200-06) was purchased from PeproTech (Rocky Hill, CT, USA). Nitrocellulose membranes and chemiluminescent horseradish peroxidase (HRP) substrate were obtained from Millipore (Billerica, MA, USA). Gefitinib was purchased from Selleckchem (Houston, TX, USA). DAB color solution, hematoxylin solution, and eosin staining solution were purchased from Servicebio (Wuhan, China).

### 4.3. Gene Reporter Assay

SKA cells were established by transfecting A549 cells with a vector containing STAT3-based luciferase reporter gene [25]. SKA cells were seeded into 96-well white plates (Corning, NY, USA) at 8000 cells/well and incubated overnight in an incubator at 37 °C with 5% CO_2_. On the second day, cells were treated with the different doses (0, 0.5, 1, 2.5, 5, 10, 25 μM) of the indicated compounds for 24 h. DMSO was used as a control. After 10 μL of stable firefly luciferase substrate (Promega, Beijing, China) was added to each well, the plates were incubated in the dark for 10 min. Luciferase activities were measured by a SpectraMax^®^ L microplate reader (Molecular Devices, Madison, WI, USA), and data were thereby obtained. 

### 4.4. Molecular Docking

Molecular docking was performed using MOE (Molecular Operating Environment) with AMBER10: EHT forcefield. The STAT3 crystal structure (PDB: 1BG1) used for docking was selected and downloaded from the Protein Data Bank (PDB, http://www.rcsb.org, accessed on 10 December 2021). The induced-fit docking approach was applied with consideration of the side-chain flexibility of residues at the binding site. The best scored conformation with minimum binding energy from the 20 docking conformations of the ligands was selected for analysis.

### 4.5. Surface Plasmon Resonance (SPR) Analysis

A purified STAT3 protein (10 μg/mL) was injected onto a CM5 chip (GE Healthcare, Chicago, IL, USA) for immobilization. Then, different concentrations (1.5625, 3.125, 6.25, 12.5, 25 μM) of compound **25** dissolved in running buffer (filtered 1 × PBS, 0.01% DMSO) were passed through the chip to generate response signals. The binding affinity (KD) was evaluated using Biacore Insight Evaluation Software T200.

### 4.6. Cellular Thermal Shift Assay (CETSA)

A549 cells were cultured in 10 cm dishes (Corning, NY, USA) and treated with 15 μM of compound **25** or vehicle (DMSO) for 2 h the next day. The cells were collected and washed twice with PBS (Servicebio, Wuhan, China), then collected in 1 mL of PBS with 1% protease and phosphatase inhibitors and dispensed into 0.2 mL PCR tubes. Each tube was heated for 3 min at the indicated temperature and cooled to room temperature, then immediately frozen in liquid nitrogen to lyse cells through three freeze–thaw cycles. Cell lysate samples were centrifuged at 20,000× *g* for 20 min at 4 °C and boiled with loading buffer for 5 min at 95 °C. Finally, the samples were analyzed by Western blot.

### 4.7. Western Blot

A549 and DU145 cells were seeded into 6-well plates (Corning, NY, USA) overnight, then incubated with different concentrations of **25** for 2 h, to detect upstream protein levels and phosphorylated STAT3 Y705; and for 24 h, to detect the expression of downstream proteins. Drug-treated cells were collected, washed twice with PBS, lysed with cell lysis buffer (RIPA cell lysis with 1% protease inhibitor and 1% phosphatase inhibitor A, B) to extract total protein, then quantified by bicinchoninic acid (BCA) assay, and denatured in a metal bath at 95 °C for 5 min. Lysates were loaded onto 10% SDS-PAGE gels for separation and transferred to nitrocellulose membranes. Then, the nitrocellulose membranes were sealed with 5% skim milk powder for 1 h and incubated with relevant primary and secondary antibodies. Finally, the corresponding target proteins were detected with chemiluminescence HRP substrate (Millipore, MA, USA) and photographed using the Tanon 5200 chemiluminescence imaging system (Biotanon, Shanghai, China).

### 4.8. IL-6 Induction of STAT3 Phosphorylation

Hela and MDA-MB-231 cells were seeded in 6-well plates and adhered overnight. The following day, cells were treated with the indicated doses (0, 2.5, 5, 10, 15 μM) of **25** for 2 h and stimulated with IL-6 (20 ng/mL) for 20 min. The untreated and unstimulated cells served as blank control. Cells were lysed and tested by immunoblotting.

### 4.9. Cell Viability Assay

Tumor cells (3000–6000/well) were plated into 96-well plates (Corning, NY, USA), and the different concentrations (0, 0.1, 0.25, 0.5, 1, 2.5, 5, 10, 25, 50 μM) of compounds were added the next day. After drug treatment for 72 h, 10 μL of resazurin (1 mg/mL) (Sigma-Aldrich, St. Louis, MO, USA) was added to each well and incubated at 37 °C in the dark for 3–4 h. The absorbance was detected using a SpectraMax^®^ i3 (Molecular Devices, Madison, WI, USA) with a 595 nm emission wavelength and a 549 nm excitation wavelength.

### 4.10. Colony Formation

Tumor cells were seeded into 6-well plates for 24 h, with 800–1000 cells/well, and treated with compound **25** at 0, 1, 2.5, or 5 μM for 1–2 weeks. When visible clones appeared on the plates, the colonies were fixed with 4% paraformaldehyde fix solution (Beyotime, Beijing, China) and stained with 0.2% crystal violet (Beyotime, Beijing, China). After the stains were washed and dried, images were photographed, and then processed using Photoshop. Colonies were quantified using ImageJ V1.8.0.

### 4.11. Wound Healing Assay

A549 and DU145 cells were separately seeded into 6-well plates. At 80% cell growth, the cells were scratched with pipette tips in each well. Previous medium was discarded, the cells were gently washed with PBS, and fresh medium was then added containing the specified concentrations (0, 5, 15 μM) of **25**. After incubation for 48 h, changes in scratch width were observed and photographed using a Zeiss Axio Vert.A1 inverted microscope.

### 4.12. In Vivo Studies

Animal experiments were approved by the Animal Policy and Ethics Committee of the Ocean University of China (ID Number: OUC-SMP-2020-11-01). Six-week-old male BALB/c mice (SPF degree, 18–22 g weight, nu/nu) were purchased from Beijing Vital River Laboratory Animal Technology Co., Ltd. (Beijing, China). Approximately 5 × 10^6^ human A549 cells were injected subcutaneously into each nude mouse. When tumor volume reached nearly 100 mm^3^, tumor-bearing mice were randomly separated into four groups (7 mice/group): a vehicle group (90% normal saline, 10% DMSO), a gefitinib group (100 mg/kg), and two compound **25** groups (5 or 10 mg/kg). The vehicle group and the compound **25** groups were intraperitoneally injected every 2 days for 2 weeks. The gefitinib group received intragastric administration every 2 days for 2 weeks. Body weight and tumor volume were recorded every 3 days. At the end of the trial, mice were euthanized with CO_2_, and tumor masses were excised for weighing and photographing.

### 4.13. Immunohistochemistry (IHC) Analyses

Mouse tissue samples were collected, fixed in 4% PFA, embedded in paraffin, and cut into slices. Then, the sections were deparaffinized in xylene, rehydrated in graded ethanol, boiled in antigen retrieval solution, and incubated with fresh 3% H_2_O_2_. Next, the slides were blocked with nonfat dry milk and incubated with the primary antibody (Ki67 1:500, p-STAT3 1:100, c-Myc 1:1600) at 4 °C overnight, followed by incubation with the HRP-conjugated secondary antibody (1:200) at room temperature (Boster, Wuhan, China), and finally added dropwise with DAB color solution (DAB dilution solution and 50 × DAB stock solution, 50:1). For hematoxylin and eosin staining (H&E staining), the tissue sections were incubated in hematoxylin solution (hematoxylin staining solution, hematoxylin fractionation solution, hematoxylin returning blue solution) and then counterstained with eosin staining solution. Images were taken with an upright fluorescence microscope (Olympus BX53, Tokyo, Japan).

### 4.14. Data Analysis and Statistical Methods

Statistical analysis was performed by GraphPad Prism 8.0 software. One-way analysis of variance (ANOVA) determined the differences between control and experimental groups. Data were expressed as specified in each case in the figure legends. *p*-Values < 0.05 were considered significant.

## Figures and Tables

**Figure 1 ijms-24-08859-f001:**
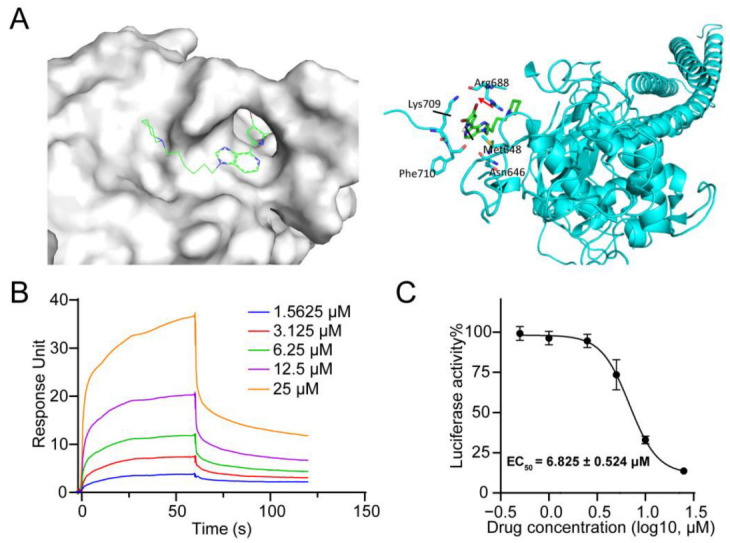
The interaction of compound **25** with the STAT3 protein. (**A**) The docking result of compound **25** with STAT3. (**B**) The affinity of compound **25** binding to the STAT3 protein was analyzed by SPR. (**C**) SKA cells were treated for 24 h with varying concentrations (0, 0.5, 1, 2.5, 5, 10, 25 μM) of compound **25** to detect luciferase activity. Experiments were performed independently at least three times, and data are expressed as mean ± SD.

**Figure 2 ijms-24-08859-f002:**
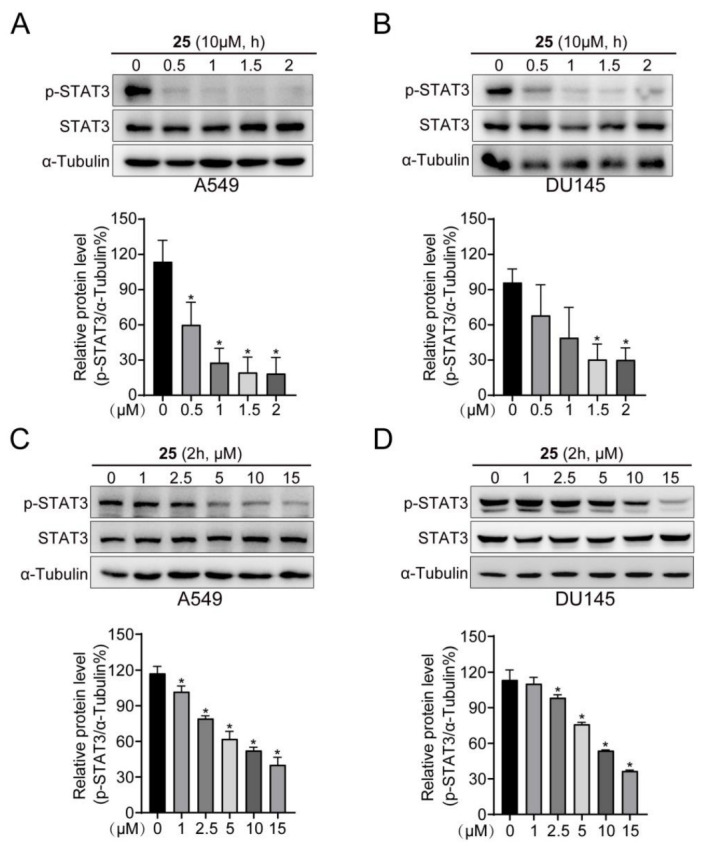
Compound **25** inhibited constitutive STAT3 Y705 phosphorylation. (**A**,**B**) A549 (**A**) and DU145 (**B**) cells were treated with 10 μM compound **25** for different times (0, 0.5, 1, 1.5, 2 h), and then lysed to detect the expression of pY705-STAT3 by Western blot. Results are expressed as mean ± SD. *n* = 3. * *p* < 0.05 vs. control. (**C**,**D**) A549 (**C**) and DU145 (**D**) cells were treated with different concentrations (0, 1, 2.5, 5, 10, 15 μM) of compound **25** for 2 h, and the expression of pY705-STAT3 was examined by Western blot. Results are expressed as mean ± SD. *n* = 3. * *p* < 0.05 vs. control.

**Figure 3 ijms-24-08859-f003:**
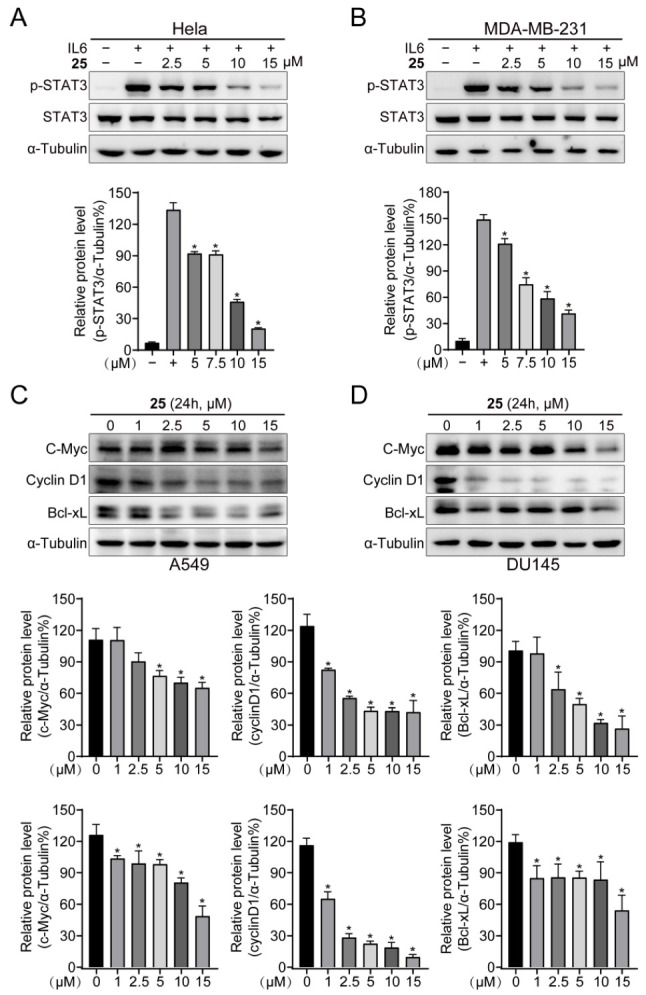
Compound **25** inhibited IL-6-induced pY705-STAT3 and the expression of downstream proteins. (**A**,**B**) Hela (**A**) and MDA-MB-231 (**B**) cells were treated with compound **25** at 0, 2.5, 5, 10, 15 μM for 2 h, then stimulated with IL-6 (20 ng/mL) for 20 min. The expression of STAT3 Y705 phosphorylation upon IL-6 stimulation was detected by Western blotting. Results are expressed as mean ± SD. *n* = 3. * *p* < 0.05 vs. control. (**C**,**D**) A549 (**C**) and DU145 (**D**) cells were treated with compound **25** at 0, 1, 2.5, 5, 10, 15 μM for 24 h, and then lysed for Western blot analysis. The expression of the downstream gene proteins c-Myc, Cyclin D1, and Bcl-xL was detected. Results are expressed as mean ± SD. *n* = 3. * *p* < 0.05 vs. control.

**Figure 4 ijms-24-08859-f004:**
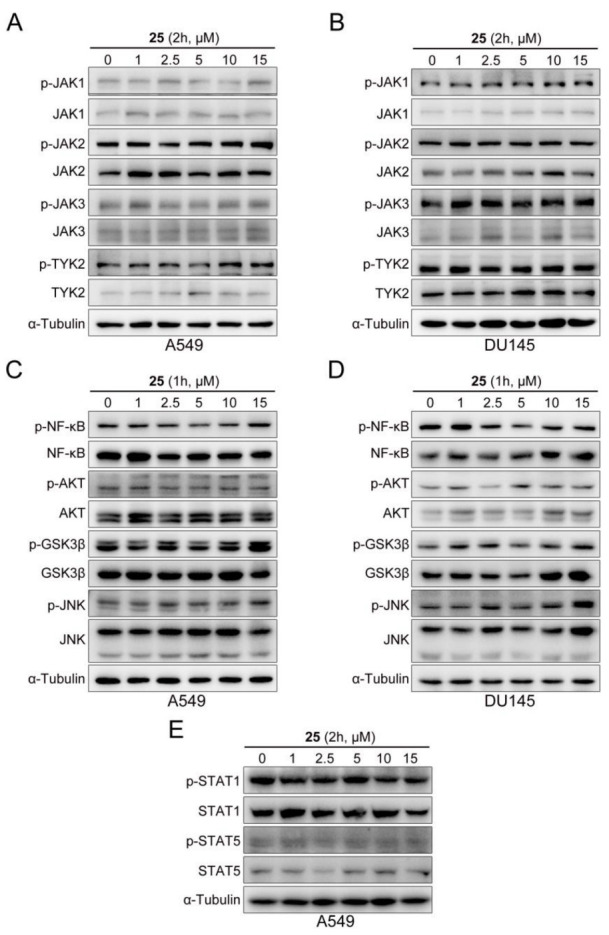
Effects of compound **25** on p-JAK, other signaling pathways, and other STAT. (**A**,**B**) A549 (**A**) and DU145 (**B**) cells were treated with different concentrations (0, 1, 2.5, 5, 10, 15 μM) of compound **25** for 2 h. Subsequently, protein lysates were collected, and the expression of p-JAK was detected by Western blot. *n* = 3. (**C**,**D**) A549 (**C**) and DU145 (**D**) cells were treated with different concentrations (0, 1, 2.5, 5, 10, 15 μM) of compound **25** for 1 h. The expressions of p-NF-κB, p-AKT, p-GSK3β, and p-JNK were detected by Western blot. *n* = 3. (**E**) A549 cells were treated with different concentrations (0, 1, 2.5, 5, 10, 15 μM) of compound **25** for 2 h. The expressions of p-STAT1 and p-STAT5 were detected by Western blot. *n* = 3.

**Figure 5 ijms-24-08859-f005:**
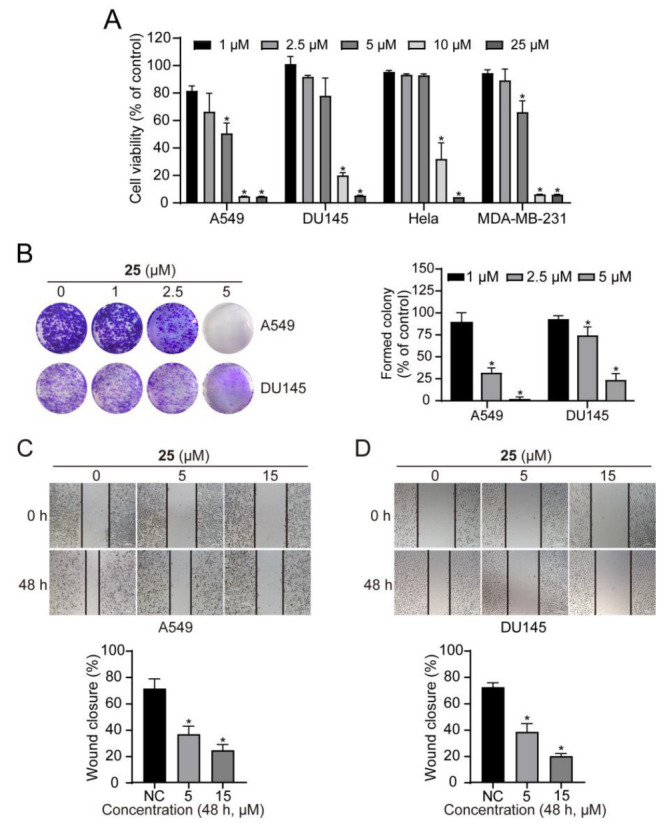
Compound **25** inhibited the proliferation, survival, and migration of cancer cells. (**A**) Cancer cells were treated with compound **25** (0, 1, 2.5, 5, 10, 25 μM) for 72 h, and cell viability was assayed by resazurin. Data are expressed as mean ± SD. *n* = 3. * *p* < 0.05 vs. control. (**B**) A549 and DU145 cells were treated with compound **25** at 0, 1, 2.5, 5 μM for 1–2 weeks. When colonies were visualized, they were stained with crystal violet. Data are expressed as mean ± SD. *n* = 3. * *p* < 0.05 vs. control. (**C**,**D**) A549 (**C**) and DU145 (**D**) cells were scratched with pipette tips and then treated with compound **25** at 0, 5, 15 μM for 48 h. Data are expressed as mean ± SD. *n* = 3. * *p* < 0.05 vs. control.

**Figure 6 ijms-24-08859-f006:**
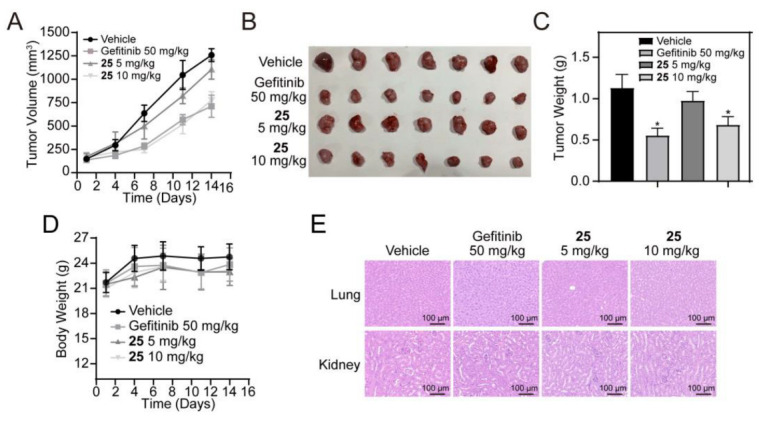
Compound **25** suppressed the growth of an A549 xenograft tumor model in vivo. Nu/nu mice were injected subcutaneously with A549 cells and treated with vehicle (i.p. normal saline with 10% DMSO/2d), compound **25** (i.p. 5 or 10 mg/kg/2d), or gefitinib (i.g. 50 mg/kg/2d) for 2 weeks. (**A**) Tumor volumes of A549 xenograft tumors in each group were measured every three days. Tumor volumes were calculated by the formula: length × width^2^ × 0.50. Values are expressed as mean ± SD. *n* = 7. (**B**) The tumors in each group of mice were excised and photographed. (**C**) Tumor weights in each group were measured after excision. Data are expressed as mean ± SD. *n* = 7. * *p* < 0.05 vs. vehicle. (**D**) Mice weights in each group were measured every 3 days during treatment. Data are expressed as mean ± SD. *n* = 7. (**E**) H&E staining of liver and kidney tissues in each group of mice. Scale bar = 100 μm.

**Figure 7 ijms-24-08859-f007:**
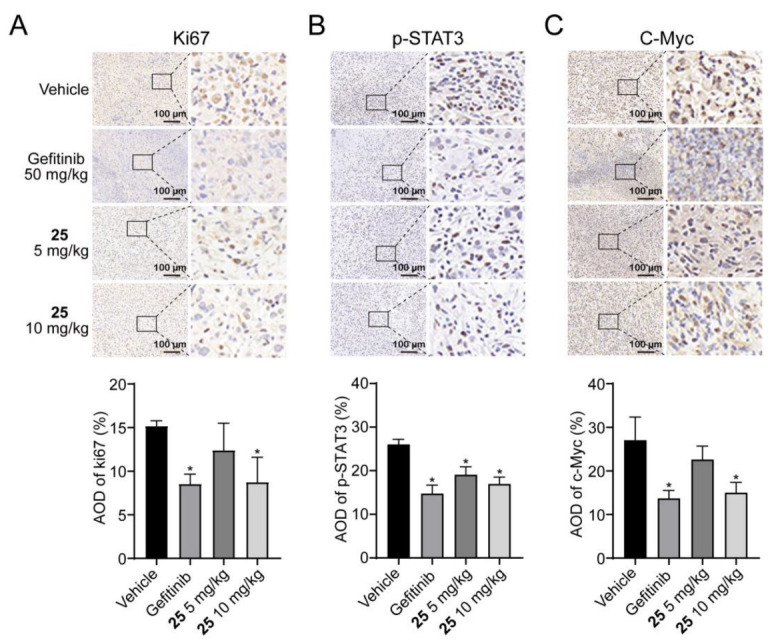
IHC for the protein levels of Ki67, pY705-STAT3, and c-Myc in the tumor tissues. (**A**) Ki67, (**B**) pY705-STAT3, (**C**) c-Myc. Scale bar = 100 μm. Data are expressed as mean ± SD. *n* = 3. * *p* < 0.05 vs. vehicle.

**Table 1 ijms-24-08859-t001:** Effects of ageladine A derivatives on the STAT3 signaling pathway.

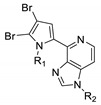
Compound	R_1_	R_2_	Salt Form	Log2 (Positive Control/Negative Control) ^a^
**14**	CH_3_	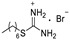	/	−11.28
**15**	H	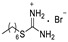	HCl	−10.99
**17**	H	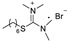	HCl	−1.88
**19**	H	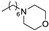	CF_3_COOH	−8.91
**25**	H	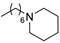	CF_3_COOH	−12.72
**27**	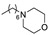	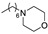	/	−2.90
**28**	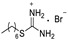	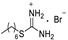	/	−11.85

^a^ The inhibitory strength of compounds on the STAT3-dependent reporter system.

**Table 2 ijms-24-08859-t002:** Antiproliferative activity of compounds **14**, **15**, **25**, and **28** on different cancer cell lines.

Compound	IC_50_ ± SD (μM) ^a^			
	A549	DU145	HELA	MDA-MB-231
**14**	22.63 ± 1.59	>25	>25	19.48 ± 0.63
**15**	10.30 ± 0.82	16.69 ± 1.97	18.579 ± 0.85	14.72 ± 2.05
**25**	4.42 ± 0.42	8.73 ± 1.53	8.665 ± 0.34	5.60 ± 1.36
**28**	>25	>25	>25	>25

^a^ The inhibitory effects of compounds on the proliferation of the four cell lines were measured by the MTT method. All experiments were performed independently at least three times, and data are expressed as means ± SD.

**Table 3 ijms-24-08859-t003:** Antiproliferative activity of compound **25** on normal cell lines.

Compound	IC_50_ ± SD (μM) ^a^	
	HUVEC	BM
**25**	31.62 ± 0.83	>50

^a^ The inhibitory effect of compound **25** on the proliferation of the four cell lines was measured by the MTT method. All experiments were performed independently at least three times, and data are expressed as means ± SD.

## Data Availability

The authors declare that the data supporting the findings of this study are available within the article and its Appendix A.

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
