# Peer review of "A Novel Ageladine A Derivative Acts as a STAT3 Inhibitor and Exhibits Potential Antitumor Effects"

_ijms, 2023, doi:10.3390/ijms24108859_

Round 1
Reviewer 1 Report
This paper describes the development of new STAT3 inhibitors from a natural compound known as Ageladine A. They start with a high number of potential molecules and narrow down their search with a compound that presents good results in different cell and animal models, where the compounds decrease the number of tumoral cells or the size of the tumors in vivo. Although it is unclear whether the antitumoral properties are directly caused by inhibition of STAT3 -the authors do not really prove it-, it is possible that this mechanism is involved, as STAT3 regulates cancer cell proliferation and survival. However, this work has a series of major caveats that need to be addressed, indicated below.
While the article is well organized and the results and conclusions are understood, the article must be proof-read by a professional proofreader, as there are many expressions and terms that are incorrectly used. There are unnecessary or redundant paragraphs and sentences overtly ambiguous, all of it to be avoided in professional scientific writing. I pointed out some in the attached file, but the article must be revised by a native speaker with experience in scientific writing.
The organization of the sections is also a little immature and inconsistent through the manuscript. Section 2.1 (Chemistry) should not be a section at all, especially when it will be only one line of text, as all the results are shown as supplementary. Just start results saying that the compounds were synthesized as described in suppl. fig X, and then move on in the same paragraph to describe how a few compounds were selected for their effect on STAT3 transcriptional activity. Furthermore, while sections 2.1, 2.2, and 2.2.1 have as titles the Methods/assays employed (immature writing, similar to a internal lab report), the rest of the section titles describe the conclusions (as it should be in a professional article). And so on. In conclusion, the article needs deep revision of writing style and English.
Major issues:
1. From a technical point of view, the methods are not well described, missing relevant information for reproducibility such as concentrations or compositions of buffers and reagents: Lysis buffers, permeabilization solution, concentrations of PI and annexin V, the concentrations of antibodies used in immunoprecipitation,... they are only a few examples of this lack of specificity in the description of methods that make the techniques difficult to be reproduced by others. But there are many other issues: How many cells did they analyze in flow cytometry analyses? There is a minimum to consider results representative or significant. How many experiments did they carry out in western blotting experiments? We should know how reproducible are those results. etc, etc, etc...When did they use the standard error and when the standard deviation in their graphs? Some of these are indicated in the attached file, but Methods and Figure Legends should be thoroughly revised beyond my comments.
2. Microscopy results in Figure 4 are not really convincing, and even less when combined with the strange results from flow cytometry data (Fig. 6E). First, cells do not look happy in Fig. 4A. There are only 2 cells in each picture, surrounded by a lot of empty space, and these cells have a very small nucleus and are very rounded even for HEK cells. It seems they are dying, and it is difficult not to see that increasing concentrations of the compound seems to kill the cells.
The co-localization studies should be carried out properly, there is software for this, but in this case there is total overlap of the signals, which makes it difficult to determine whether they really colocalize (=interact) or are simply homogeneously distributed in the same space (the whole cell in this case). So, with these images as a starting point, there is no chance that the best co-localization software can really work accurately. Better pictures should be provided or this part of the experiment should be completely removed from the article.
The same happens with Fig. 4C, where green signal is totally unfocused, and it is difficult to read the images and get a conclusion out of them. Merging blue and green channels do not add anything to the pictures, but it is obvious that there is nuclear signal in all cases, and therefore the compound does not really prevent STAT3 from going to the nucleus. Also, if the authors want to conclude that pSTAT3 is reduced by the compound, it is not clear in this image. Fluorescence levels in at least 150 cells from 3 independent experiments all taken with the same exposure and light intensity should be quantified and statistics done on it. However, this is better proven by means of western blots with the corresponding phosphorylated antibodies, as they do in Figure 3. Furthermore, it is unclear whether cells in Fig. 4C were stimulated or not with cytokines. Does pSTAT3 signal correspond to constitutive activation of STAT3 in these cells or to inducible phosphorylation in response to cytokines? But I would remove all fluorescence microscopy pictures from the article, as they are not convincing nor add anything reliable.
3. Flow cytometry analysis also has serious flaws. Fig. 6E. Cell cycle analysis should show a clear difference between G0/G1 and G2/M peaks, and an S phase clearly distinguished between them. Please, check the literature for this sort of profile, how it should look. The profiles shown are from a badly done cell cycle protocol or a machine without the necessary sensitivity, as they only show one large peak, hardly differentiating G1 from S phases, and no G2/M peak. Without the proper cell cycle distribution pattern, all analysis and modelling of the cell cycle is flawed and therefore useless. These experiments should be repeated and proper cell cycle patterns shown.
Fig. 6F. is also wrong from several points of view. First, it seems that the compound 25 is autofluorescent, since when it is present, the whole population of cells becomes positive in both FL1 (Annexin V) and FL2 (PI) channels, and cells become more fluorescent in both channels in a dose dependent manner. Therefore, changes in fluorescence are actually dubious or inconclusive, not necessarily due to PI or annexin V stainings (=cell death), and not demonstrating cell death. In A549 cells, there is a possible increase in both PI and annexin V stainings, but I don´t think this happens in DU145. These results are not convincing nor conclusive because these issues are not properly addressed. Second, the two detectors are clearly not compensated for intrusions of signal in each of them, a basic issue that should have been considered.
Considering all these issues, I don´t think the authors have the necessary expertise in flow cytometry to present and analyze this sort of data in a professional way, as these issues are obvious to any professional user.
4. Western blotting. There was a time when we could show one western blot and pass revision without criticisms. However, western blotting is a source of major misdeeds, and reviewers and readers have grown suspicious on this sort of results. As a consequence, it is common to present not one blot, but 3 or more blots or a graph with the statistics coming from 3-5 blots. All western blots in this article should be repeated at least 3 times FROM INDEPENDENT EXPERIMENTS, and the bands quantified and represented in a graph with error bars and statistics. It is true that in some cases the authors show results in different cell lines, which is reassuring, but they should repeat the experiments for each western blot. This is especially true for results that are unclear to say the least, such as fig. 7D, where it is unclear whether pSTAT3 is actually decreased by compound 25, unless we consider ratios between pSTAT3 and total STAT3, which are not presented. However, to be honest, I don´t think you will find an actual decrease after you do the ratios in this WB in particular.
5. Other missing quantifications: Figs. 6B and 7E. In 6B, it is mentioned somewhere that the colonies were quantified, with n=2, but this experiments should be repeated at least one more time, and the quantifications and errors properly represented and analyzed. In Figure 7E, the number of positive cells should also be quantified in at least 3 independent experiments, and the data represented with their errors included, to have an idea of the reproducibility and variability of the system.
6. There are parts that are lengthy, wordy and useless to the manuscript, such as the discussion paragraph in lines 286-306, which is basically the introduction all over again, without connection to the results.

Round 2
Reviewer 1 Report
I appreciate you removed all conflicting results, but my suggestions were to correct those results, not to simply removed them. Immunofluorescence images and flow cytometry analysis with selected concentrations and time points, once you know more or less the outcome, should be straight-forward, and if you added them in the first place was because they could actually provide relevant information. Removing them just makes the article poorer and does not address my concerns.
Additionally, I requested extensive English editing by a professional proof-reader, and this clearly did not happen. However, this time I am not going to go through the manuscript again.
Round 3
Reviewer 1 Report
Thank you for the explanation, I understand your difficulties. The article reads better with the new proof-reading, but it seems MDPI services do not cover figures: the X axis of the graphs in Figures 5C and 5D say "Concertration", when they should say "Concentration".
Author Response
Thank you very much for your careful reading. We feel very sorry for our careless mistakes. We have modified "Concertration" to "Concentration" for the X-axis of the graphs in Figure 5C and 5D. Thanks again for your correction.
In addition, we have double checked our manuscript and graphs. Any revisions to the manuscript have been marked using the “Track Changes” function. A clean version of the revised manuscript has also been prepared for convenience.
Thank you again for all your time involved and for giving us this great opportunity to improve our manuscript. We hope that you will find this revised version satisfactory.